# Using web log analysis to evaluate healthcare students' engagement behaviours with multimedia lectures on YouTube

Cailbhe Doherty [1,2]*

**1** School of Public Health, Physiotherapy and Sports Science, University College Dublin, Dublin, Ireland,
**2** Insight Centre for Data Analytics, University College Dublin, Dublin, Ireland

* cailbhe.doherty@ucd.ie

**Data Availability Statement:** The data for this investigation have been made shareable and can be accessed on OSF.io using the following link: https://osf.io/9aerk/.

## Abstract

The objective of this study was to utilise web log analysis to evaluate the relationship between University students' engagement (e.g., watch time) and the characteristics of a catalogue of multimedia lectures, including their duration, the speaking rate of the narrator and the extent to which they implemented certain principles from Mayer's Cognitive Theory of Multimedia Learning (CTML). Fifty-six multimedia lectures covering topics related to healthcare (e.g., anatomy, physiology and clinical assessment) were developed to differentially employ the image/embodiment, redundancy, segmentation and signalling principles from the CTML. These lectures were delivered to multiple cohorts of students throughout an academic semester. Student watch time was evaluated using the meta-usage data provided by YouTube studio. The multimedia lectures were viewed 4338 times (mean = 35 views per lecture; 27 unique viewers per lecture). Generalised estimating equations revealed that videos that were segmented into shorter chunks, that incorporated signals to highlight important information for students and during which captions were toggled 'off' by students were associated with longer watch times (P < 0.05). Additionally, watch time diminished for videos placed later in a sequence based on the audience retention metric. When designing multimedia lectures, instructors should be encouraged to use on screen labels to highlight important information, segment learning material into shorter 'chunks' and incorporate a dynamic instructor on screen at regular intervals displaying high embodiment. If several videos are to be delivered to students as part of a learning 'unit', educators should consider placing the most important learning material earlier in the sequence.

## Introduction

Healthcare students are increasingly turning to online sources for medical information with audiences for legacy media decreasing [1]. Today, students in medicine, nursing, radiography and physiotherapy have at their disposal a catalogue of audio, video, text and mixed reality-based media on different content sharing platforms to support their learning needs [2]. Coinciding with increasing demands, the online community has produced and shared a vast library of free multimedia content for healthcare students (e.g., [3]). The video-sharing site YouTube

**Funding:** The author received no specific funding for this work.

**Competing interests:** The author has declared that no competing interests exist.

is particularly popular for sharing this content. YouTube is the largest video-sharing website [4], with over 2.1 billion monthly active users [5]. Given YouTube's wide reach, there has been an increasing interest from educators in using this platform for disseminating learning materials [6]. The small but growing body of empirical evidence guiding the design and delivery of learning material through YouTube reflects this increasing interest [7, 8].

Video sharing platforms like YouTube are by nature "pluralistic, participatory and social" [9], presenting unique opportunities for teaching staff to engage the student body in new ways. While a number of strands of research have sought to investigate how healthcare students interact with web-based content, insights have been constrained by the design of the experiment (which is often cross-sectional [10]), the data acquisition methodologies (usually interviews and questionnaires about behaviour, rather than the behaviour itself [11]) or the framing of the research question within a preferred model of cognition and learning [12].

In contrast to interviews, surveys or think-aloud exercises, transaction log analysis—a broad categorisation of methods including Web log analysis, blog analysis, and search log analysis—does not involve directly interfacing with participants to elicit their web-browsing behaviours [13]. A transaction log is an electronic record of interactions that have occurred between a system and the users of that system [13]. Research using transaction log analysis of YouTube videos has traditionally focused on popularity metrics such as views, comments and likes [14–16]. Building on this line of work, YouTube studio, an auxiliary of the main YouTube site, offers more nuanced measures of user engagement like average view duration, otherwise known as "audience retention", which reveal closer details about how well a video holds users' attention. To date however, universities have been slow to fully utilise the potential of web log analysis tools like those provided by YouTube studio to evaluate their students' engagement with multimedia video lectures. While many universities are capturing meta-usage data from across their own systems like e-resource usage and computer logins [17], important insights are likely being missed about their audiences' retention patterns. These insights might help us to understand how students interact with instructional multimedia online and could inform educators about how best to develop, design and deliver multimedia content on web-based platforms to support learning.

Good design and pedagogy likely predicate engagement with online content [18, 19], however what represents 'good design' for healthcare multimedia remains unclear. Frameworks like the Cognitive Theory of Multimedia Learning (CTML) posit that certain principles should be followed to maximise knowledge transfer. For instance, the "signalling principle advocates the use of screen labels to highlight important material for the learner [20–22] while the "segmenting principle" dictates that learning material should be split learning into shorter chunks to manage cognitive load [23]. Frameworks like the CTML offer a way to identify the determinants of engagement with audio, video, text-based and mixed-reality healthcare multimedia on platforms such as YouTube.

Therefore, the aim of this study was to conduct a transaction log analysis to evaluate the relationship between video lecture design and user engagement. To do so, a catalogue of multimedia videos was developed and uploaded to YouTube. This catalogue of video lectures employed a variety of design principles [24] and web log analysis was then combined with survey data and quizzes to evaluate user engagement with each multimedia lecture.

Our hypotheses were as follows:

1. Multimedia lectures that follow certain design principles form the CTML [25] will be associated with higher levels of engagement (i.e., audience retention)

2. Multimedia lectures that follow certain design principles from the CTML [25] will be associated with better knowledge retention (i.e., scores on a quiz)

3. Students will rate multimedia lectures that follow certain design principles as more supportive to their learning.

## Methods

Due to the exploratory nature of this research, the protocol was not pre-registered. The study was approved by the ethics committee of the institution at which the primary author was based (*ref*: LS-20-47).

### Experimental design

Web log analysis [13] was used to evaluate learners' engagement behaviours during multimedia instruction. Web log analysis is a form of transaction log analysis [13]. Transaction log analysis enables macro-analysis of aggregate user data. A transaction log is an electronic record of interactions that have occurred between a system and users of that system, for example, clicks on a website or specific web link, number of page views or watch time on a video. In this study, we used the web logs provided by YouTube Studio [26] and Brightspace (D2L Europe Ltd) to evaluate students' transaction behaviours during multimedia instruction. These outcomes are described in further detail below.

### Study site & student cohort

The study site was the University at which the primary author is based. The learning material that was used for web log analysis included a catalogue of multimedia video lectures and covered topics relevant to the undergraduate and post-graduate degree programs in physiotherapy. Briefly, learning material included lectures concerning anatomy, physiology, electrotherapeutic agents, clinical assessment, ethics in research, information governance and evidence based practice. The multimedia lectures were delivered in the form of 'learning units' to students enrolled in program modules. Prior to the start of the semester, students were informed that their engagement with the learning material for the module would be evaluated as part of a research study; they were informed that data were only being gathered and analysed at the group level and that no individual personal or identifiable information would be used. The need for consent was waived by the ethics committee because no identifiable data about individual participants was captured (only aggregate data).

### Multimedia lecture catalogue

A catalogue of 56 multimedia lectures were included in this web log analysis. The development of multimedia lectures for each topic and lecture followed a series of steps. First, a lecture script and accompanying storyboard were prepared for each topic. A teleprompter system was setup, and an instructor was recorded as they read the script aloud. Audio input was recorded via a microphone (Yeti, Blue Designs Inc, CA, USA) connected to a laptop and video input was recorded (1080p at 25 frames per second) using a Canon 250d DSLR camera (Canon Inc, Tokyo, Japan). Following this, accompanying videos and slides were developed based on the original storyboard. Both the audio and video media were then imported into a commercially available video editing software (Final Cut Pro; Apple Inc, CA, USA) and time-aligned. The audio and video recordings were trimmed to remove mistakes (e.g. mispronunciations) and large gaps. Standardised logos, colours, formatting and sequencing were applied over the video media to establish a recognisable profile for the multimedia lectures [18]. Finally, captions were created for each video. These were 'off' by default but could be toggled 'on' if viewers chose (detailed later).

During the design and development of the multimedia lectures, selected principles from the CTML [25] were incorporated, to various extents, throughout each video. Specifically, for the 'signalling' principle, on screen labels were used in varying amounts within each multimedia lecture to highlight important material [27]. For the 'image' and 'embodiment' principles, a video of the instructor enthusiastically explaining relevant topics or theories was presented intermittently throughout each lecture [28]. Lectures with higher levels of signalling or image/ embodiment had more labels or clips of the instructor interspersed throughout, respectively. For example, the range of time during which labels were present on screen in the catalogue of multimedia lectures was between 7–87% of their overall duration. The range of time during which an instructor was present on screen in the catalogue of multimedia lectures was between 0–60% of their overall duration.

For the 'segmenting' principle, lectures were split into 'chunks' of different duration [29]. Lectures that were split into shorter chunks were deemed to represent 'higher' levels of segmenting. The shortest multimedia lecture was 57 seconds in duration and the longest was 19 minutes and 21 seconds in duration.

For the 'redundancy' principle [30], whether students toggled captions 'off' or 'on' was recorded using YouTube Studio; if captions were toggled on, this was considered to represent greater redundancy of information as the captions always contained the same information being presented visually or aurally. Captions were toggled by the viewer cohort between 0–87% of the time.

In summary, the development processes resulted in the creation of a catalogue of fifty-six multimedia lectures which incorporated each of the selected multimedia instructional principles on a continuous scale. A visual depiction of how these principles were followed are displayed in Fig 1.

## Capture of web log data

All multimedia lectures were exported to YouTube (YouTube Inc, CA, USA) with 'private' visibility (i.e., not accessible via the open web). During the week of term that the lecture was scheduled to be delivered to students, the visibility for that lecture was altered to 'unlisted' and students were provided with a link to view the multimedia lecture for a 1-week period as part of a learning 'unit' via the university's virtual learning environment (VLE), Brightspace (D2L Europe Ltd). 'Unlisted' videos on YouTube can only be access by those with a specific link and are not searchable.

The learning unit for each multimedia lecture consisted of the video for that week's topic, a topic-specific multiple choice questionnaire (MCQ) and a single-item Likert survey where students were asked to rate their agreement with the statement "This lecture enhanced my learning/understanding of course content" on a scale from 1 ("strongly disagree") to 5 ("strongly agree"). MCQs and surveys were only accessible in the topic week, after which they were made unavailable. The quizzes that accompanied each video were comprised of between 3–5 questions covering the material that was presented in that lecture. None of students' behaviours, their responses on the Likert survey or their scores on the MCQs contributed towards their overall grade in the module.

Web log data were categorised into three domains of engagement: 'affective' engagement was determined via: 1) the number of submissions on the Likert-style survey that accompanied each video and; 2) the median score for each survey; 'behavioural' engagement was determined via the audience retention which is defined by YouTube studio as the average duration that a video is watched and is expressed as a % of its overall length; 'cognitive' engagement was determined via: 1) the number of students who attempted the quiz that

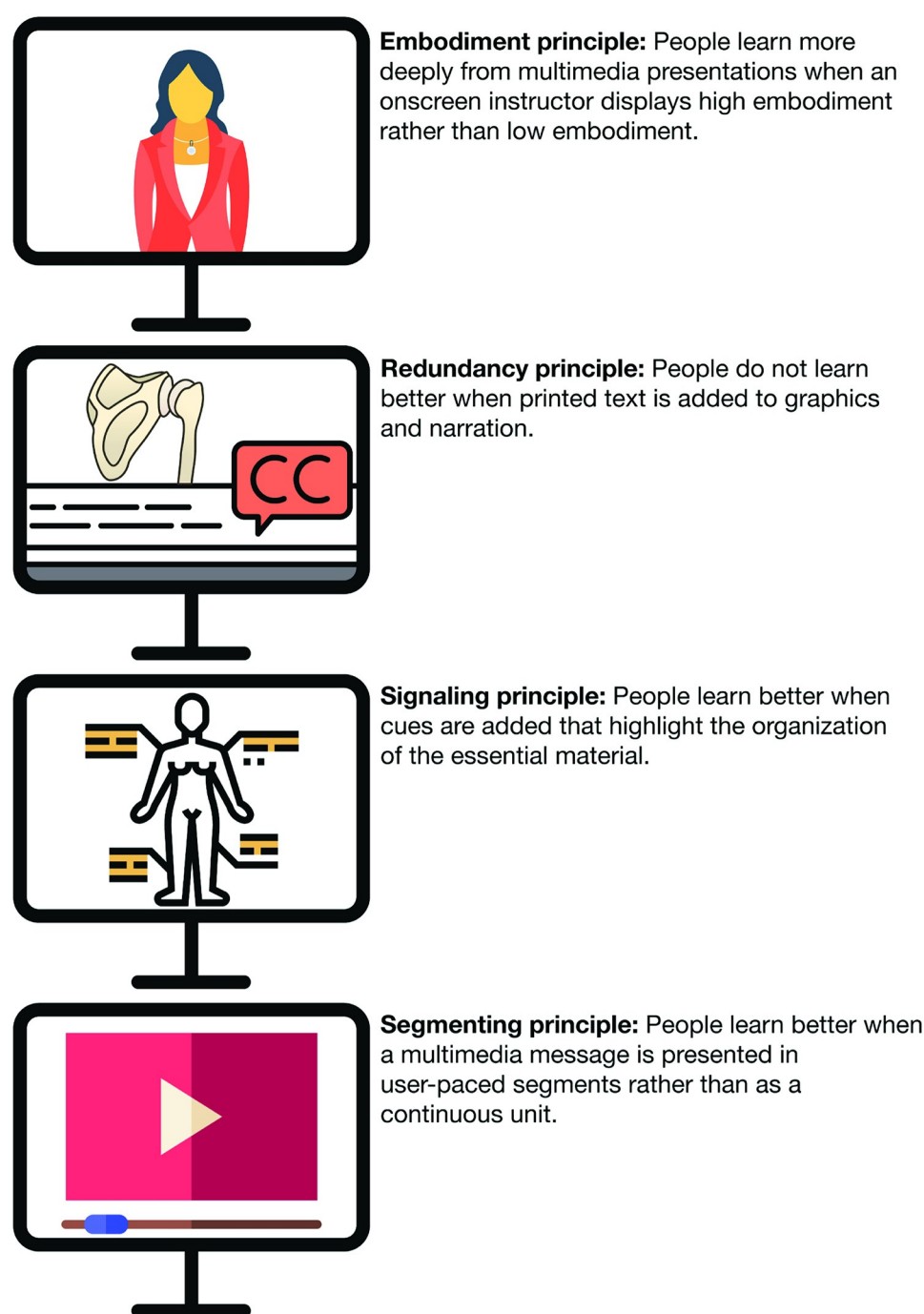

**Fig 1. Demonstration and explanation of the image/embodiment, redundancy, signalling and segmentation principles.**

accompanied each video and; 2) the average score for each quiz. The number of survey submissions and quiz attempts and the median and average scores associated with each, respectively, was logged using the Brightspace VLE. Audience retention was logged using YouTube studio analytics.

## Outcomes and data analysis

The characteristics of the multimedia lectures were considered to represent selected principles from the CTML and were expressed using the following variables: 1) duration (segmenting principle); 2) its placement in the sequence of videos students were required to watch for that week; 3) the percentage of the overall video duration during which on screen labels or signals were used to highlight important information for the viewer (signalling principle); 4) the percentage of the overall video duration during which a presenter was displayed on screen (image/embodiment principle); 5) the speaking rate of the narrator, expressed as the number of words spoken per minute; finally, whether captions were toggled 'on' by the students during the video (redundancy principle) was expressed as a percentage of the number of views accrued by each video. Descriptive statistics were used to summarize multimedia lecture characteristics and aggregate-level engagement behaviours with these lectures. Continuous variables are expressed as means or medians, and standard deviations (sd). The experimental outcomes are defined in Table 1.

Generalised estimating equations (GEE) were used to evaluate the relationship between the design characteristics of multimedia lectures and measures of engagement. Each of the characteristics described above (duration, sequence, speaking rate and the percentage of time during the lecture where signals and a "talking head" of the lecturer were incorporated) were included as covariates in separate GEE models for each measure of engagement. The number of submissions and the number of attempts on the surveys and quizzes, respectively, were expressed as a percentage of the number of students who had access to the videos. An exchangeable correlation structure was used, and each model was corrected for dependent observations by including the multimedia lecture code as a subject effect. The *a priori* p value for this analysis was set at $p < 0.05$.

## Data access

Data collected during the current experiment can be accessed using the following link to open access repository, OSF.io:

**Table 1. The complete list of and definitions of the experimental outcomes, including the multimedia characteristics and measures of engagement.**

| | | |
|---|---|---|
| Multimedia characteristics | Image/ Embodiment | The amount of time the lecturer was presented on screen, enthusiastically demonstrating learning material. Expressed as a percentage of overall multimedia lecture duration. |
| | Redundancy | The percentage of students who toggled subtitles 'on' as they watched the multimedia lecture. Captions represent redundancy of information because they words spoken by a narrator. |
| | Segmenting | The length of the multimedia lecture (in seconds). |
| | Sequence | The placement of a multimedia lecture in the sequence of videos students were required to watch for that week |
| | Signalling | The amount of time signals, defined as on screen visual cues (such as arrows, or short pieces of text), were present on screen. Expressed as a percentage of overall multimedia lecture duration. |
| Cognitive engagement | Quiz attempts | The number of attempts completed by learners on the quiz associated with a multimedia lecture. Expressed as a percentage of the number of learners who had access to the quiz (i.e. the total number of learners who had access to the learning units [N = 92]) |
| | Quiz scores | The average score achieved by learners on a quizzes that accompanied each multimedia lecture. |
| Behavioural engagement | Audience retention | The average view duration for a multimedia lecture, expressed as a percentage of overall multimedia lecture length. |
| Affective engagement | Survey submissions | The number of submissions completed by students on the Likert survey associated with a multimedia lecture. Expressed as a percentage of the number of students who had access to the survey (i.e. the total number of learners who had access to the learning units [N = 92]) |
| | Median survey score | The median score on a Likert-survey that accompanied each multimedia lecture in which students were asked to respond to the statement "This lecture enhanced my learning/understanding of course content" on a scale from 1 ("strongly disagree") to 5 ("strongly agree"). |

Note: audience retention values can exceed 100% when learners watch a multimedia lecture (or parts of a multimedia lecture) multiple times.

[removed for the purposes of anonymisation]

## Results

During the observation period, the catalogue of 56 multimedia lectures were collectively viewed 4338 times. These videos were watched by different cohorts of students (enrolled in different modules) between 2019 and 2022; 19/56 lectures were accessed by two student cohorts; 16/56 lectures were access by three student cohorts; 12/56 lectures were access by four student cohorts and 9/56 lectures were access by five student cohorts. As a result the 56 multimedia lectures were associated with 124 access periods, each lasting 1-week. The mean number of views per video during these access periods was 35, with 27 unique viewers per video.

The average view duration was 89.1% of the overall multimedia lecture length (range 34% to 142%). There were a total of 3285 quiz attempts (an average of 26% of the cohort of learners attempted the quiz that accompanied each multimedia lecture) and the average score on the quizzes was 70.2%. There were 1743 survey submissions (an average of 14% of the cohort of learners completed the survey that accompanied each multimedia lecture) and the median score on the Likert-survey that accompanied each multimedia lecture (in which students were asked to respond to the statement "This lecture enhanced my learning/understanding of course content" on a scale from 1 ["strongly disagree"] to 5 ["strongly agree"]) was 4. For the 124 access events, technical difficulties with the VLE meant that students did not have access to six surveys throughout the observation period and as a result, no data were collected for these (4.8% missing data for this outcome).

The characteristic of the multimedia lectures (i.e., duration, sequence, speaking rate, use of captions and the percentage of time during the lecture where signals and a "talking head" of the lecturer were incorporated) are presented in Table 2. Descriptive statistics of learners' engagement behaviours with the multimedia lectures (including the number of survey submissions, the median survey score, the audience retention, the number of quiz attempts and the average quiz scores for each multimedia lecture) can be accessed using the following link to the open access repository, OSF.io:

[removed for the purposes of anonymisation]

### Association between multimedia lecture design with affective, behavioural and cognitive engagement behaviours

**Affective engagement.** The GEE for the number of submissions on the Likert style survey estimated a main effect for segmenting (p = 0.004), signalling (p = 0.005) and image/embodiment (incorporation of a "talking head" in the multimedia lecture) (p = 0.007). An analysis of the parameter estimates associated with these main effects revealed that there were a greater number of survey submissions when multimedia lectures were shorter, and when signals and image/embodiment were incorporated to a greater extent within them.

The GEE for the median score on the Likert style survey estimated no main effects (P > 0.05).

**Behavioural engagement.** The GEE for audience retention estimated a main effect for sequence (p = 0.003), segmenting (p < 0.0005), signalling (p = 0.038) and for the use of captions (p = 0.008). There was also a significant interaction between sequence and duration (p < 0.0005). An analysis of the parameter estimates associated with these main effects revealed that audience retention was higher for shorter videos placed earlier in a sequence that incorporated more signals and where captions were not used.

**Cognitive engagement.** The GEE for the quiz attempts estimated a main effect for segmenting (p = 0.021), speaking rate (p = 0.046) and signals (p = 0.008). An analysis of the

**Table 2. Characteristics of video lecture catalogue.**

| Title | Sequence | Duration (HH:MM:SS) | Signals | Image/ Embodiment | Captions |
|---|---|---|---|---|---|
| Consumer wearable devices part 1 (introduction) | 4 | 00:05:18 | 7% | 14% | 12% |
| Consumer wearable devices part 2 (the research evidence) | 5 | 00:13:36 | 20% | 38% | 57% |
| Ethics & information governance part 1 (introduction) | 1 | 00:04:30 | 31% | 25% | 40% |
| Ethics & information governance part 2 (the ethics committee) | 2 | 00:05:06 | 45% | 41% | 40% |
| Ethics & information governance part 3 (informed consent) | 3 | 00:08:04 | 15% | 38% | 33% |
| Ethics & information governance part 4 (data management) | 4 | 00:08:32 | 32% | 25% | 33% |
| Evaluating muscle weakness part 1 (introduction) | 1 | 00:03:09 | 37% | 46% | 71% |
| Evaluating muscle weakness part 2 (assessment) | 2 | 00:07:40 | 36% | 32% | 29% |
| Functional anatomy part 1 (assessment) | 2 | 00:05:56 | 31% | 44% | 22% |
| Functional anatomy part 1 (introduction) | 1 | 00:04:50 | 62% | 44% | 19% |
| Goniometry part 1 (introduction) | 3 | 00:04:56 | 72% | 16% | 30% |
| Goniometry part 2 (assessment) | 4 | 00:03:41 | 66% | 25% | 25% |
| Human gait part 1 (introduction) | 1 | 00:03:48 | 43% | 20% | 0% |
| Human gait part 2 (functional anatomy) | 2 | 00:04:41 | 46% | 12% | 0% |
| Human gait part 3 (kinematics) | 3 | 00:05:32 | 52% | 13% | 0% |
| Human posture part 1 (introduction) | 3 | 00:07:06 | 13% | 38% | 18% |
| Human posture part 2 (assessment rationale) | 4 | 00:04:35 | 30% | 60% | 79% |
| Human posture part 3 (assessment method) | 5 | 00:05:22 | 21% | 40% | 24% |
| Innate immunity part 1 (introduction) | 1 | 00:03:09 | 51% | 39% | 84% |
| Innate immunity part 2 (inflammation) | 2 | 00:02:43 | 25% | 45% | 82% |
| Innate immunity part 3 (components of innate system) | 3 | 00:04:21 | 28% | 37% | 80% |
| Innate immunity part 4 (remodelling & repair) | 4 | 00:04:31 | 32% | 45% | 0% |
| Introduction to pain physiology part 1 (noxious stimuli) | 5 | 00:05:29 | 38% | 42% | 86% |
| Introduction to pain physiology part 2 (reflex arcs) | 6 | 00:03:37 | 35% | 41% | 78% |
| Introduction to pain physiology part 3 (the placebo effect) | 7 | 00:03:37 | 25% | 19% | 70% |
| Low Level Laser Therapy | 1 | 00:13:27 | 62% | 32% | 70% |
| Neuromuscular Electrical Stimulation (NMES) | 1 | 00:09:26 | 61% | 14% | 81% |
| Reading research: a beginner's guide | 1 | 00:19:21 | 59% | 31% | 6% |
| The ankle complex (articular structures) | 2 | 00:02:14 | 66% | 3% | 13% |
| The ankle complex (muscular structures) | 3 | 00:04:01 | 80% | 0% | 29% |
| The ankle complex (osseous structures) | 1 | 00:03:25 | 66% | 4% | 29% |
| The brachial plexus | 1 | 00:03:52 | 46% | 5% | 33% |
| The elbow, wrist & hand (articular structures) | 2 | 00:05:04 | 84% | 21% | 0% |
| The elbow, wrist & hand (muscular structures) | 3 | 00:13:13 | 69% | 19% | 0% |
| The elbow, wrist & hand (osseous structures) | 1 | 00:08:21 | 85% | 16% | 0% |
| The foot complex (articular structures) | 2 | 00:03:34 | 87% | 1% | 32% |
| The foot complex (muscular structures) | 3 | 00:06:11 | 78% | 1% | 26% |
| The foot complex (osseous structures) | 1 | 00:02:08 | 47% | 8% | 20% |
| The hip complex (articular structures) | 2 | 00:00:57 | 51% | 0% | 27% |
| The hip complex (muscular structures) | 3 | 00:05:32 | 22% | 0% | 33% |
| The hip complex (osseous structures) | 1 | 00:03:56 | 53% | 0% | 26% |
| The knee complex (articular structures) | 2 | 00:04:17 | 68% | 14% | 32% |
| The knee complex (muscular structures) | 3 | 00:06:20 | 53% | 4% | 34% |
| The knee complex (osseous structures) | 1 | 00:03:36 | 68% | 3% | 32% |
| The lumbosacral plexus | 2 | 00:06:14 | 20% | 5% | 31% |
| The principles of evidence based medicine | 1 | 00:13:04 | 33% | 34% | 6% |
| The shoulder complex (articular structures) | 2 | 00:02:49 | 78% | 12% | 0% |

*(Continued)*

**Table 2.** (Continued)

| Title | Sequence | Duration (HH:MM:SS) | Signals | Image/ Embodiment | Captions |
|---|---|---|---|---|---|
| The shoulder complex (muscular structures) | 3 | 00:13:58 | 71% | 3% | 0% |
| The shoulder complex (osseous structures) | 1 | 00:04:54 | 65% | 10% | 0% |
| Therapeutic Ultrasound | 1 | 00:06:34 | 49% | 26% | 80% |
| Thermotherapy part 1 (ice) | 1 | 00:12:47 | 53% | 17% | 74% |
| Thermotherapy part 2 (heat) | 2 | 00:04:56 | 55% | 20% | 77% |
| Transcutaneous Electric Nerve Stimulation | 1 | 00:10:02 | 59% | 28% | 20% |
| Wearable devices part 1 (introduction) | 1 | 00:05:39 | 20% | 29% | 72% |
| Wearable devices part 2 (design) | 2 | 00:03:29 | 63% | 6% | 67% |
| Wearable devices part 3 (evidence) | 3 | 00:06:55 | 19% | 41% | 67% |

Quiz attempts and survey submissions are expressed as a percentage of the overall number of students who had access to the learning unit.

parameter estimates associated with these main effects revealed that a greater number of students attempted the quiz that accompanied each multimedia lecture when videos were shorter, when the speaking rate was faster and when signals were incorporated to a greater extent within the multimedia lectures.

The GEE for the average score on the quiz associated with each multimedia lecture revealed no main effects for lecture style ($p > 0.05$).

## Discussion

The aim of this study was to evaluate how the design of multimedia lectures covering learning material relevant to a university-level healthcare program can influence students' engagement with them. The learning material was part of the course syllabi in undergraduate and postgraduate degree programs in physiotherapy and included video lectures on anatomy, physiology, electrotherapeutic agents, clinical assessment, ethics in research, information governance and evidence based practice. Lectures were delivered to students over the course of three academic years between September 2019 and 2022. In this period, the 56 videos were incorporated in 12 separate modules; they were viewed 4338 times.

We hypothesized that multimedia lectures which incorporated certain design principles outlined in Mayer's CTML, including the signalling, segmenting, redundancy, image and embodiment principles, would be more engaging to students [31]. Specifically, our primary hypothesis was that each principle would be independently associated with higher levels of audience retention, and could be 'stacked' together in a single video to further hold students' attention. Our findings partially confirmed our primary hypothesis as only the segmenting, signalling and redundancy principles increased audience retention: videos that were segmented into shorter chunks, that incorporated signals to highlight important information for students and during which captions were toggled 'off' by students (thereby reducing the amount of redundant information on screen), were associated with longer watch times, relative to their overall length. Contrary with our primary hypothesis, no effect was found for the image/embodiment principle on audience retention.

Our secondary hypothesis was that each principle would increase 'affective' and 'cognitive' engagement, both independently and when stacked together. Affective and cognitive engagement were each evaluated both by the number of submissions and attempts on the surveys and quizzes that accompanied each video in addition to the scores on those surveys and quizzes. Our results showed that while the segmenting, signalling and image/embodiment principles

increased 'affective' engagement, based on the number of survey submissions, only the segmenting and signalling principles increased 'cognitive' engagement based on the number of quiz attempts; contrary to our secondary hypothesis, none of the principles evaluated had any effect on survey or quiz scores.

While not discussed in the framework of the CTML, we also sought to explore the influence of the narrator's speaking rate and whether the placement of the video in a sequence might play a role in overall engagement, each of which have previously been linked with engagement in student learning [23]. The placement of a video lecture in a sequence is analogous to the segmenting principle in that it is a kind of 'sectional arrangement' where videos are organised into a meaningful cluster for the student. According to the CTML, segmenting videos in this way (as opposed to delivering material in one continuous chunk) can be used to manage essential cognitive processing for learners [29], however the principle does not specifically concern behavioural engagement as defined in this study, how attention may wane over the course of several videos placed in a sequence. Our findings showed that, while speaking rate had no effect on any of our experimental outcomes (in contrast to prior research [23]), average watch time did indeed diminish for videos placed later in a sequence based on the audience retention metric. We also observed an interaction effect between sequence and duration, suggesting that the negative effect on audience retention is compounded if longer videos are placed later in a sequence.

Taken together, these results support the development of multimedia lectures in healthcare education that use on screen labels to highlight important information for the learner (i.e., the signalling principle [27]), that segment learning material into shorter 'chunks' (i.e., the segmenting principle [23]), that incorporate a dynamic instructor on screen at regular intervals (i.e., the image principle [32]) displaying high embodiment (i.e., the embodiment principle [28]) to increase students' watch time and their engagement with supplementary learning material like quizzes and surveys. Furthermore, if multiple videos are to be delivered to students covering a specific topic or as part of a learning 'unit' together, these results advocate either placing the most important learning material earlier in the sequence or perhaps splitting material up further so as to allow students a kind of 'cognitive respite' before they progress to the next video in a sequence [33].

It's important to note that the measures of affective, behavioural and cognitive engagement that were used in this study measure different behavioural constructs than the three "goals of multimedia instruction" in the CTML, which are to manage essential cognitive processing, reduce extraneous cognitive processing and foster generative cognitive processing [34]. Briefly, extraneous cognitive processing is cognitive processing that does not serve the instructional goal, it is related to surplus content or confusing design. Essential cognitive processing is the cognitive processing required to represent the essential material in working memory, it is related to the complexity of the learning material and its overall quantity (i.e., its length or duration). Generative cognitive processing is cognitive processing required for deeper understanding, it is related to the learner's motivation to learn. The CTML posits that each of its fifteen principles can only be used to achieve one of the three cognitive processing goals [25]; the segmenting, pre-training and modality principles are used to manage essential cognitive processing; the coherence, signalling, redundancy and contiguity principles are used to reduce extraneous cognitive processing; the personalisation, voice, embodiment, multimedia and generative activity principles are used to reduce extraneous cognitive processing. Taking an example from the catalogue of lectures used in the current study, if a healthcare student was presented with an illustration of an immune cell, the components of which were described through words spoken by an on-screen instructor, they would have to engage in extraneous cognitive processing to discern which key terms were linked with the different parts of the

illustration; on-screen labels could be used to reduce assist this process. Similarly, if the instructor demonstrated high embodiment, using an expressive voice, and gesticulating naturally, the learner may exert more effort to understand what the instructor is presenting.

On the basis of the CTML framework, each principle *should* have independently increased "knowledge retention" (based on the quiz scores), however no single principle was associated with this outcome in our study. And while the signalling and segmenting principles increased all measures of engagement, it is interesting that the principles of image and embodiment, which for the purposes of this study were combined because of their inherent commonality, only increased students' generative activity (i.e., their motivation to complete the feedback survey that accompanied each video). These findings are at odds with some of the studies investigating the CTML [35–38] and demonstrates that there isn't a simple causal link between principle and cognitive processing goal, that certain principles may play larger (or smaller) effects on cognitive processing and that these effects may be context-specific. For instance, a study investigating the embodiment principle might compare knowledge transfer in two groups of students using a narrated slideshow in which an instructor stands next to the presentation while speaking. In a "low-embodied version", the instructor would stand still throughout the lecture with only her lips moving. In the "high-embodied" version, the instructor would gesture with her hands as she talked, shift her body position from time to time, make natural facial expressions, and move her eyes towards the learner [39]. It is in these contexts, under this kind of study design, which has previously linked each CTML principle with improvements in knowledge transfer among research participants. However, these contexts and methodologies do not qualify as "real-world evidence" [40]; the learning units typically only exist as experimental entities with no real-world relevance to either the control or intervention groups. So while several meta-analyses have shown the effectiveness of signalling [35], segmenting [36], redundancy [41] and embodiment [37] for improving knowledge retention based on test performance, to the author's knowledge this has yet to be replicated in real-world contexts. One prior study we conducted in a similar setting, although over a shorter time period in a significantly smaller cohort of students, similarly found no association between the signalling, segmenting and embodiment principles and test performance [6].

There are a number of possible explanations for this. First, the catalogue of multimedia lectures included in the current study were developed, primarily to fulfil the learning outcomes of the modules in which they incorporated, and then to investigate the extent to which they incorporated the design principles of the CTML and student engagement; this experiment was ancillary to students' learning-it would have been unethical to conduct otherwise. Second, each multimedia lecture also adhered to many of the other principles from the CTML (including the multimedia, modality, personalisation, voice, coherence, spatial contiguity and temporal contiguity principles) and the catalogue did not include any lecture which did not stack at least two of the signalling, segmenting, redundancy and image/embodiment principles. Together, the combination of principles within each multimedia lecture may have compromised the protocol's ability to reveal subtle differences in students' test performance.

Similar to the analysis we previously published [6], another possible explanation for the lack of an effect on test performance is that, because the content of the entire catalogue of lectures was relevant to each student cohorts' course material, the students who completed the quizzes may have had higher levels of natural motivation to learn multimedia lecture content. Whereas an average of 42.6% of students watched each video at least once (based on the number of unique views for each video, expressed as a percentage of class size), only 26% of students completed the quiz, and the average score on the quizzes was 70.2% which corresponds to an 'A' grade (or first class honours); this high score may be representative of the highly

motivated sub-cohort of students who completed the quizzes and may not reflect the efficacy (or lack thereof) of each principle, hence why we also included the number of quiz attempts in our analysis, which revealed a positive effect for the segmenting and signalling principles.

Despite its novelty, its external validity and important practical relevance for educators in several healthcare disciplines in particular, this study is not without limitations.

As discussed, because the multimedia lecture catalogue was developed for specific health-care courses (including medicine, physiotherapy, medical informatics and health and performance science), while this controlled for the potentially confounding effects of sample heterogeneity, future research should be conducted seeking to replicate these findings in other fields. Next, while transaction log analysis is an effective approach to evaluate student behaviours [13], data can only be analysed at an aggregate level [42]. Therefore, our findings provide limited insights into individual students' behaviour or levels of engagement. Finally, this study was grounded in the CTML, and while this has provided a useful framework for mapping the reality of multimedia design and engagement to learning processes over several decades of research, it was not developed in or for real-world contexts. Students today have access to an enormous catalogue of online multimedia, which has been developed by a global community of content creators, including teachers, researchers, academics and their peers. This content invariably combines multiple principles together, uses 'seductive techniques' like music and rhetorical agents [43], is delivered in multiple formats (e.g., podcasts, full-length videos, video-shorts and blogs) across several platforms (e.g., YouTube, Instagram, TikTok, Vimeo and Spotify), all of which likely have an independent and interacting effects on the behavioural processes of how users' engage with and learn from the knowledge contained therein; the CTML may not be well-suited to evaluating this kind of content. Due to the diversity of online content, simpler meta-data (e.g., format, duration, readability, image quality, frame-rate) may provide the foundation upon which new or more practical frameworks might be developed.

In conclusion, this study provides real-world evidence of the effectiveness of the, signalling, segmenting, redundancy, image and embodiment principles for increasing the amount of time students spend watching instructional multimedia and their engagement with supplementary learning material like surveys and quizzes. When designing multimedia lectures, instructors should be encouraged to use on screen labels to highlight important information, segment learning material into shorter 'chunks' and incorporate a dynamic instructor on screen at regular intervals displaying high embodiment. If several videos are to be delivered to students as part of a learning 'unit', the most important learning material should be placed earlier in the sequence as students' attention will likely wane over multiple videos. While the CTML has provided a valuable framework for developing and evaluating multimedia in psychological experimentation contexts, it may not be well suited to appraising the vast catalogue of online multimedia on platforms such as YouTube, which typically 'stack' principles together and use seductive techniques such as music, narrative agents, branding and production techniques related to lighting, sound, scenery and props to capture users' attention; further research is required to explore the potential importance of these agents.

## Author Contributions

**Conceptualization:** Cailbhe Doherty.

**Data curation:** Cailbhe Doherty.

**Formal analysis:** Cailbhe Doherty.

**Investigation:** Cailbhe Doherty.

**Methodology:** Cailbhe Doherty.

**Project administration:** Cailbhe Doherty.

**Resources:** Cailbhe Doherty.

**Software:** Cailbhe Doherty.

**Supervision:** Cailbhe Doherty.

**Validation:** Cailbhe Doherty.

**Visualization:** Cailbhe Doherty.

**Writing – original draft:** Cailbhe Doherty.

**Writing – review & editing:** Cailbhe Doherty.

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
