## [Decision Letter · Decision Letter 0]

26 Jan 2023

PONE-D-22-29297Using web log analysis to evaluate healthcare students’ engagement behaviours with multimedia lectures on YouTube.PLOS ONE

Dear Dr. Doherty,

Thank you for submitting your manuscript to PLOS ONE. After careful consideration, we feel that it has merit but does not fully meet PLOS ONE’s publication criteria as it currently stands. Therefore, we invite you to submit a revised version of the manuscript that addresses the points raised during the review process.

We look forward to receiving your revised manuscript.

Kind regards,

Tauseef Ahmad

Academic Editor

PLOS ONE

Journal Requirements:

Additional Editor Comments:

Major revision required before further process

Reviewers' comments:

Reviewer's Responses to Questions

**Comments to the Author**

1. Is the manuscript technically sound, and do the data support the conclusions?

Reviewer #1: Yes

Reviewer #2: Yes

Reviewer #3: Yes

2. Has the statistical analysis been performed appropriately and rigorously? 

Reviewer #1: Yes

Reviewer #2: Yes

Reviewer #3: Yes

3. Have the authors made all data underlying the findings in their manuscript fully available?

Reviewer #1: Yes

Reviewer #2: Yes

Reviewer #3: Yes

4. Is the manuscript presented in an intelligible fashion and written in standard English?

Reviewer #1: Yes

Reviewer #2: Yes

Reviewer #3: Yes

5. Review Comments to the Author

Reviewer #1: The subject addressed in this investigation has considerable importance nowadays. However, at this moment, this draft seems like a minor continuation of an earlier article [1] by the same author. Thus, I cannot recommend the publication of this manuscript in its present form.

[1] Doherty, Cailbhe. "An investigation into the relationship between multimedia lecture design and learners’ engagement behaviours using web log analysis." Plos one 17.8 (2022): e0273007.

https://journals.plos.org/plosone/article?id=10.1371/journal.pone.0273007

Reviewer #2: This is an interesting study, supporting known multimedia principles to design better instructional videos for healthcare students. Also, it demanded work to prepare the YouTube videos. The major research contribution of this study is that it is more ecologically valid than the typical laboratory experiment. The study found stronger support for the segmenting, signaling, and redundancy principles, and somewhat smaller support for the image and embodiment principles. It is a timely topic, and the study is relevant for PLOS ONE. I have only minor suggestions to improve the study before accepting it for publication. My suggestions are detailed next.

MINOR

1 – Page 5 Lines 100¬-102 (P5 L100-102): Hypotheses 1 and 2 have a wrong “[ref]” that I think it should be changed to the proper reference numbers.

2 – In the Results section, be more consistent by using 3 decimal places in all reported ps. For example, P16 L283 and P17 L301 used only 2 decimal places for the p.

3 – Discussion, second paragraph could start with a reference in P17 L314. For example (see reference below), it could read: “…would be more engaging to students (see Castro-Alonso et al., 2021)”.

4 – P19 L358-359, when discussing the “cognitive respite”, a supporting reference could be used, so this could be: “…before they progress to the next video in a sequence (see the spacing effect in Chen et al., 2018), “

5 – P23 L453 is too harsh against CTML. I would revise “…the CTML is ill-suited to evaluating…” to “…the CTML may not be well suited to evaluating…”

6 – P23 L458 repeats “signalling, segmenting”.

7 – P23 L466 up to the end, is also too harsh against CTML, and it is speculative. I would delete all that last sentence, which begins with “While the CTML has provided a valuable…”.

8 – In the reference list, P26 and P27 repeat the same book, which is single-authored (not an edited book). So, references 28, 29, 30, 31, 32, 34, 35 should be only one.

Castro-Alonso, J. C., de Koning, B. B., Fiorella, L., & Paas, F. (2021). Five strategies for optimizing instructional materials: Instructor- and learner-managed cognitive load. Educational Psychology Review, 33(4), 1379-1407. https://doi.org/10.1007/s10648-021-09606-9

Chen, O., Castro-Alonso, J. C., Paas, F., & Sweller, J. (2018). Extending cognitive load theory to incorporate working memory resource depletion: Evidence from the spacing effect. Educational Psychology Review, 30(2), 483-501. https://doi.org/10.1007/s10648-017-9426-2

Reviewer #3: This manuscript is about using transaction log analysis to evaluate the relationship between video lecture design (characteristics of these multimedia lectures, including implementation of certain principles from Mayer's Cognitive Theory of Multimedia Learning (CTML)) and healthcare students' engagement.

The manuscript is technically sound, with clear objectives, rigorously conducted, and the conclusions were supported by the presented data.

Kindly find the following recommendations:

1- add a brief paragraph about the CTML to the introduction section.

2- add a reference/ references for lines (62-66, page 3).

3- What does [ref] found in lines (100 and 102, page 5) stand for?

4- add the ORCID number of the corresponding author to the title page.

6. PLOS authors have the option to publish the peer review history of their article (what does this mean?). If published, this will include your full peer review and any attached files.

Reviewer #1: No

Reviewer #2: No

Reviewer #3: **Yes: **Mohammad Issam Eddeen Abu Assab

---

## [Author Response · Author response to Decision Letter 0]

7 Feb 2023

Response to reviewers

Reviewer #1: The subject addressed in this investigation has considerable importance nowadays. However, at this moment, this draft seems like a minor continuation of an earlier article [1] by the same author. Thus, I cannot recommend the publication of this manuscript in its present form.

[1] Doherty, Cailbhe. "An investigation into the relationship between multimedia lecture design and learners’ engagement behaviours using web log analysis." Plos one 17.8 (2022): e0273007.

https://journals.plos.org/plosone/article?id=10.1371/journal.pone.0273007

AUTHOR RESPONSE: Many thanks for taking the time to review this manuscript. You are correct in asserting that the current paper is a continuation of previous work, however the dataset is significantly larger (4x), the population is more heterogenous and the multimedia lecture catalogue was more diverse. One of the main limitations with the cited paper was that it related to a single cohort of students in 1-year of higher education and related to one topic area (anatomy). The current paper overcomes these limitations by evaluating a more diverse cohort of students in various stages of both undergraduate and postgraduate degree programs across a range of modules and degree programs. That the findings from the first paper have been both replicated and expanded upon are an important contribution to current knowledge in this field.

Reviewer #2: This is an interesting study, supporting known multimedia principles to design better instructional videos for healthcare students. Also, it demanded work to prepare the YouTube videos. The major research contribution of this study is that it is more ecologically valid than the typical laboratory experiment. The study found stronger support for the segmenting, signaling, and redundancy principles, and somewhat smaller support for the image and embodiment principles. It is a timely topic, and the study is relevant for PLOS ONE. I have only minor suggestions to improve the study before accepting it for publication. My suggestions are detailed next.

AUTHOR RESPONSE: Thank you for the time you have taken to review the paper. The manuscript has been revised in line with your comments, details for which are presented below.

MINOR

1 – Page 5 Lines 100¬-102 (P5 L100-102): Hypotheses 1 and 2 have a wrong “[ref]” that I think it should be changed to the proper reference numbers.

AUTHOR RESPONSE: Amended as requested

2 – In the Results section, be more consistent by using 3 decimal places in all reported ps. For example, P16 L283 and P17 L301 used only 2 decimal places for the p.

AUTHOR RESPONSE: The items to which you refer relate to the statistical output of the GEE, the convention for which is to report tro 4 decimal places when p<0.005 and to report exact p-values to three decimal places otherwise. As such, we have not made any changes to these values, but would defer to the editor if necessary.

3 – Discussion, second paragraph could start with a reference in P17 L314. For example (see reference below), it could read: “…would be more engaging to students (see Castro-Alonso et al., 2021)”.

AUTHOR RESPONSE: Amended as requested. Thank you for the relevant citation.

4 – P19 L358-359, when discussing the “cognitive respite”, a supporting reference could be used, so this could be: “…before they progress to the next video in a sequence (see the spacing effect in Chen et al., 2018), 

AUTHOR RESPONSE: Amended as requested. Thank you for the relevant citation.

5 – P23 L453 is too harsh against CTML. I would revise “…the CTML is ill-suited to evaluating…” to “…the CTML may not be well suited to evaluating…”

AUTHOR RESPONSE: Amended as requested.

6 – P23 L458 repeats “signalling, segmenting”.

AUTHOR RESPONSE: Repetition removed.

7 – P23 L466 up to the end, is also too harsh against CTML, and it is speculative. I would delete all that last sentence, which begins with “While the CTML has provided a valuable…”.

AUTHOR RESPONSE: In this case, we would like to retain the sentiment that we need a more appropriate tool to evaluate multimedia design in the ‘age of YouTube’. As this is a segment related to ‘future research directions’, we would argue that our speculation is not misplaced, and hope that this final sentence is provocative enough to encourage discourse and follow-on research, while acknowledging the value of the CTML.

8 – In the reference list, P26 and P27 repeat the same book, which is single-authored (not an edited book). So, references 28, 29, 30, 31, 32, 34, 35 should be only one.

AUTHOR RESPONSE: Each citation refers to a specific chapter and page number from Mayer’s Cognitive Theory of Multimedia Learning. Our interpretation of the PLOS one author guidelines leads us to believe that this is the most appropriate way to cite the work—we defer to the editor in this instance and would be happy to make the suggested change if required.

Castro-Alonso, J. C., de Koning, B. B., Fiorella, L., & Paas, F. (2021). Five strategies for optimizing instructional materials: Instructor- and learner-managed cognitive load. Educational Psychology Review, 33(4), 1379-1407. https://doi.org/10.1007/s10648-021-09606-9

Chen, O., Castro-Alonso, J. C., Paas, F., & Sweller, J. (2018). Extending cognitive load theory to incorporate working memory resource depletion: Evidence from the spacing effect. Educational Psychology Review, 30(2), 483-501. https://doi.org/10.1007/s10648-017-9426-2

Reviewer #3: This manuscript is about using transaction log analysis to evaluate the relationship between video lecture design (characteristics of these multimedia lectures, including implementation of certain principles from Mayer's Cognitive Theory of Multimedia Learning (CTML)) and healthcare students' engagement.

The manuscript is technically sound, with clear objectives, rigorously conducted, and the conclusions were supported by the presented data.

AUTHOR RESPONSE: Thank you for your positive appraisal of this manuscript and the time you have taken to review it. We have amended the manuscript in line with your suggestions. Please see below for details.

Kindly find the following recommendations:

1- add a brief paragraph about the CTML to the introduction section.

AUTHOR RESPONSE: As requested, the following paragraph has been incorporated in the introduction:

Good design and pedagogy likely predicate engagement with online content (18, 19), however what represents ‘good design’ for healthcare multimedia remains unclear. Frameworks like the Cognitive Theory of Multimedia Learning (CTML) posit that certain principles should be followed to maximise knowledge transfer. For instance, the “signalling principle advocates the use of screen labels to highlight important material for the learner (20-22) while the “segmenting principle” dictates that learning material should be split learning into shorter chunks to manage cognitive load (23). Frameworks like the CTML offer a way to identify the determinants of engagement with audio, video, text-based and mixed-reality healthcare multimedia on platforms such as YouTube. 

Therefore, the aim of this study was to conduct a transaction log analysis to evaluate the relationship between video lecture design and user engagement. To do so, a catalogue of multimedia videos was developed and uploaded to YouTube. This catalogue of video lectures employed a variety of design principles (24) and web log analysis was then combined with survey data and quizzes to evaluate user engagement with each multimedia lecture.

Our hypotheses were as follows:

1. Multimedia lectures that follow certain design principles form the CTML (25) will be associated with higher levels of engagement (i.e., audience retention)

2. Multimedia lectures that follow certain design principles from the CTML (25) will be associated with better knowledge retention (i.e., scores on a quiz)

3. Students will rate multimedia lectures that follow certain design principles as more supportive to their learning.

2- add a reference/ references for lines (62-66, page 3).

AUTHOR RESPONSE: Appropriate citations have been added.

3- What does [ref] found in lines (100 and 102, page 5) stand for?

AUTHOR RESPONSE: Correct references added in each case.

4- add the ORCID number of the corresponding author to the title page.

AUTHOR RESPONSE: Requested information added.

---

## [Editor Report · Decision Letter 1]

27 Mar 2023

Using web log analysis to evaluate healthcare students’ engagement behaviours with multimedia lectures on YouTube.

PONE-D-22-29297R1

Dear Dr. Doherty,

We’re pleased to inform you that your manuscript has been judged scientifically suitable for publication and will be formally accepted for publication once it meets all outstanding technical requirements.

Kind regards,

Tauseef Ahmad

Academic Editor

PLOS ONE

Additional Editor Comments (optional):

Thank you for your sincere revisions according to the comments of the review
---

## [Editor Report · Acceptance letter]

5 Apr 2023

PONE-D-22-29297R1 

Using web log analysis to evaluate healthcare students’ engagement behaviours with multimedia lectures on YouTube. 

Dear Dr. Doherty:

I'm pleased to inform you that your manuscript has been deemed suitable for publication in PLOS ONE. Congratulations! Your manuscript is now with our production department. 

Kind regards, 

on behalf of

Dr. Tauseef Ahmad 

Academic Editor

PLOS ONE